# Distribution of Core Root Microbiota of Tibetan Hulless Barley along an Altitudinal and Geographical Gradient in the Tibetan Plateau

**DOI:** 10.3390/microorganisms10091737

**Published:** 2022-08-29

**Authors:** Na Wei, Xiaofeng Yue

**Affiliations:** 1Institutions of Agricultural Product Quality Standard and Testing Research, Tibet Academy of Agricultural and Animal Husbandry Sciences, Lhasa 850032, China; 2Oil Crops Research Institute of the Chinese Academy of Agricultural Sciences, Wuhan 430062, China

**Keywords:** bacterial community, elevational distribution, fungal community, microbial diversity, metabarcoding

## Abstract

The Tibetan Plateau is regarded as the third pole of the earth and is one of the least explored places on the planet. Tibetan hull-less barley (*Hordeum vulgare* L. var. *nudum*) is the only cereal crop grown widely in the Tibetan Plateau as a staple food. Extensive and long-term cropping of barley may influence the soil’s chemical and biological properties, including microbial communities. However, microbiota associated with hull-less barley is largely unexplored. This study aimed to reveal the composition and diversity of bacterial and fungal communities associated with the hull-less barley at different elevations in the Tibetan Plateau. The core bacterial and fungal taxa of Tibetan hull-less barley were identified, with Bacillaceae, Blastocatellaceae, Comamonadaceae, Gemmatimonadaceae, Planococcaceae, Pyrinomonadaceae, Sphingomonadaceae, and Nitrospiraceae being the most abundant bacterial taxa and Ceratobasidiaceae, Chaetomiaceae, Cladosporiaceae, Didymellaceae, Entolomataceae, Microascaceae, Mortierellaceae, and Nectriaceae being the most abundant fungal taxa (relative abundance > 1%). Both bacterial and fungal diversities of hull-less barley were affected by altitude and soil properties such as total carbon, total nitrogen, and available phosphorus and potassium. Both bacterial and fungal diversities showed a significant negative correlation with altitude, indicating that the lower elevations provide a conducive environment for the survival and maintenance of hull-less barley-associated microbiota. Our results also suggest that the high altitude-specific microbial taxa may play an important role in the adaptation of the hull-less barley to the earth’s third pole.

## 1. Introduction

Barley, with its two subspecies hull-less barley (*Hordeum vulgare* L.) and naked barley (*Hordeum vulgare* var. *nudum* Hooker f.), is the fourth largest cereal crop cultivated worldwide [1]. Tibetan hull-less barley (*Hordeum vulgare* L. var. *nudum*), also known as “Qingke” in Chinese, is a naked barley (i.e., the inner and outer glumes are separated and seeds are exposed) that is a member of the Gramineae family in the genus *Hordeum* [2,3]. Tibetan hull-less barley is primarily grown on the Tibetan Plateau and was one of the earliest crops known to humans (cultivated on the Tibetan Plateau ~3600 years ago) [4]. The Tibetan Plateau, also known as the “roof of the world”, has harsh environmental conditions with dangerously high levels of ultraviolet radiation, low oxygen levels, low temperatures, and low precipitation [5]. Barley, however, has strong resilience to extreme environmental conditions and is characterized by high cold tolerance with a short maturity period. Tibetan hull-less barley is the only grain that can grow and mature naturally in alpine regions with an altitude of more than 4200 m above sea level [3]. Barley accounts for about 70% of the grain production in Tibet, and it is widely used for animal feed, food, and brewing [3,6].

The rhizosphere’s microbial communities, also known as the “second genome” of plants, are the most important component of plant–microbiota relationships that maintain plant growth and health [7,8,9]. It is estimated that the microbial population in the plant root zone is as high as 10^6^–10^9^ cells/cm^3^ and interacts closely with plants [10,11]. Rhizosphere microbial communities play an important role in plant nutrient acquisition, growth, tolerance to environmental stress, and defense against pathogens and diseases [12,13]. However, microbes in the plant rhizosphere can be influenced by soil physicochemical properties and plant root exudates [11,14,15,16]. Indeed, plants can shape their rhizosphere microbiome by stimulating or inhibiting the secretion of specific microbiome enrichment [10]. 

Hull-less barley may evolve with different rhizosphere microbial communities (both beneficial and harmful) during its lifetime, including germination, maturation, and flowering [17,18]. Therefore, identifying the core root microbial community composition of hull-less barley and the influencing factors is important for barley breeding and agricultural practice. Few comprehensive studies have recently been reported on the rhizosphere microbiome of barley from the Tibetan Plateau [19]. However, past studies primarily focused on the hull-less barley’s genome, continuous cropping practices, and soil nutrients (N, P, and K) uptake by barley roots [20,21]. For example, a previous study found that the continuous cropping of hull-less barley alters the bacterial community structure and function. This could result in a significant decline in barley yields and microbial diversity, as well as increased abundance of nitrate-reduction-associated pathways [5,21]. Liu et al. [22] found that the root of hull-less barley could select bacterial and fungal communities from the bulk soil compartments, and the specific enriched microbes could promote the growth of hull-less barley and biological control against pathogens [13,22]. However, research is lacking on the altitudinal and geographical distribution that affects microbiota of hull-less-barley-associated microbiota and their relationships. With the development of high-throughput sequencing technology, the rhizosphere microbiome has been extensively studied in model plants (e.g., *Arabidopsis* sp.), including some crops (e.g., rice, wheat, and maize) [23,24,25]. Despite the significance of root microbiota for crop production and the Tibetan hull-less barley being the most important food crop in Tibet, the rhizosphere microbiome of the barley across different regions of the Tibetan Plateau remains largely unknown.

The structure and distribution of microbial communities are strongly associated with climatic conditions across different geographical locations [26,27,28]. The altitudinal span of the Tibetan Plateau forms different climatic zones with local-scale variation in temperature and moisture. Altitudinal variation and substantial differences in environmental conditions may result in the contrasting structure of microbial communities [29,30]. Hence, a better understanding of the rhizosphere microbiome of Tibetan hull-less barley in different regions of the Tibetan Plateau and their relationships with different environmental factors would provide an important direction for barley cultivation. In this study, using high-throughput amplicon sequencing, we estimated the diversity of hull-less barley’s rhizosphere bacterial and fungal communities across the altitudinal gradient of the Tibetan Plateau (ranged from 2643 m to 3806 m). The specific objectives of this study were to (1) identify the core bacterial and fungal taxa associated with hull-less barley in different geographic locations in the Tibetan Plateau; (2) determine the environmental factors that shape the structure and diversity of bacterial and fungal communities along the elevation gradient.

## 2. Materials and Methods

### 2.1. Sample Collection and Analysis of Soil Chemical and Metabolite Properties

In this study, we selected hull-less barley grown in six different geographical locations of Tibet, including Bomi (95°36′53.1″, 30°0′9.14″), Gongga (90°35′49.049″, 29°10′22.79″), Kanuo (97°4′52.80″, 30°53′26.41″), Pulan (81°9′47.16″, 30°19′28.92″), and Linzhou (91°8′54.06″, 29°57′36.97″). All the sampling sites were selected based on elevational gradients and the cultivated area of hull-less barley, spanning an altitudinal range from 2643 m to 3806 m above sea level. All the sampling sites have been used for planting hull-less barley for at least five years. When rhizosphere samples were collected, the surface soil (approximately 5 cm of topsoil) was removed, and the roots of hull-less barley were dug out. The roots were manually shaken to remove the loose soil, and the remaining ~1 mm layer of rhizosphere soil was placed in phosphate-buffered saline (PBS) in a 50 mL falcon tube. Five rhizosphere samples were randomly taken from each sampling point using the standard collection methods of rhizosphere samples as described by Edwards et al. [9]. The soil samples (approximately 5 cm of topsoil was removed) were collected from four corners and the center point of the targeted plant and analyzed for the soil’s chemical properties. The sample was placed in a 50 mL sterile test tube and transported back to the laboratory in an icebox within 24 h of sample collection. The soil samples used to analyze soil chemical properties were stored at 4 °C, and the rhizosphere samples used for DNA extraction were stored at −80 °C. The soil samples were dried and crushed at 55 °C before passing through a 2 mm sieve to remove rocks and visible plant tissue. Soil chemical properties (total carbon (TC), available phosphorus (AP), total nitrogen (TN), available potassium (AK), iron ions, and sulfur ions) were measured using a standard soil testing procedure (http://vdb3.soil.csdb.cn accessed on 19 June 2022). Briefly, TN and TC were measured using the vario TOC analyzer (Elementar, Hanau, Germany). The plant available phosphorus and potassium and iron and sulfur ions were measured using ICP-MS (Thermo Fisher, Walthamm, MA, USA). 

### 2.2. Extraction of DNA from Root Microbiome Samples

The rhizosphere microbial genomic DNA was extracted from 0.5 g soil samples with Fast DNA^®^ Spin Kit for soil using the FastPrep instrument (MP Biomedicals, Santa Ana, CA, USA) following the method provided by the manufacturer. The quality of genomic DNA was examined on 1% agarose gel electrophoresis, and the concentration and purity of DNA were determined using a NanoDrop 2000 UV-Vis spectrophotometer (Thermo Scientific, Wilmington, DE, USA). Genomic DNA samples were stored at −80 °C until use. For each sample, ITS1F (5′-CTTGGTCATTTAGAGGAAGTAA-3′) and ITS2R (5′-CTTGGTCATTTAGAGGAAGTAA-3′) were used for the amplification of the fungal ITS rRNA gene, and the primer pairs 338F (5′-ACTCCTACGGGAGGCAGCAG-3′) and 806R (5′-GGACTACHVGGGTWTCTAAT-3′) were used to amplify the highly variable region V3-V4 of the bacterial 16S rRNA gene [31]. The PCR conditions for the bacterial community were as follows: 95 °C for 3 min; 30 cycles of 95 °C for 30 s, 56 °C for 30 s, and 72 °C for 3 min; 72 °C for 10 min; hold at 4 °C. The PCR conditions for the fungal community were as follows: 95 °C for 3 min; 30 cycles of 95 °C for 30 s, 55 °C for 30 s, and 72 °C for 3 min; 72 °C for 10 min; hold at 4 °C. The PCR amplification was performed in triplicate for each sample to minimize the stochastic effect. The PCR products were purified using the AxyPrep DNA Gel Extraction Kit (Axygen Biotech, Hangzhou, China) following the manufacturer’s instructions and pooled in equimolar concentration to construct the amplicon library. The paired-end sequencing (2 × 250) was performed on the Illumina MiSeq PE300 platform at the BGI-Shenzhen Co. Ltd. (Shenzhen, China).

### 2.3. 16S rRNA Gene and ITS Region Bioinformatics and Phylogenetic Analyses

All the sequences in FASTQ format underwent quality filtering using the *fastq-filter* command in USEARCH 11.0, and the sequences with base average sequencing accuracy ≥99% were retained. The pair-end sequences were merged and dereplicated using the *fastq_mergepairs* script in VSEARCH. Then, all the chimeric sequences were detected and removed by combining de novo and reference-based methods and using the *UCHIME* command and *UNOISE3* command in USEARCH 11.0. Briefly, bacterial sequences were compared against the Ribosomal Database Project Gold (RDP Gold) database as a reference for the removal of bacterial chimera [32], and fungal sequences were compared against the UNITE UCHIME database for the removal of fungal chimera [33]. The remaining nonchimeric sequences were dereplicated and assembled into Amplicon sequence variants (ASVs), and singletons (<8 sequences) were eliminated using the *UNOISE3* command in USEARCH 11 with a 100% similarity. The representative sequences of all fungal ASVs were compared to UNITE database using a Bayesian classification algorithm, and species annotation information was obtained for each fungal ASV with 80% confidence [33]. Species annotation information for each bacterial ASV was obtained using the SILVA 16S rRNA database (release 138) with 80% confidence [34]. Species annotation was performed on all the representative sequences, and the sequences that could not be annotated to any known taxonomic unit were designed as unidentified ASVs.

### 2.4. Diversity and Statistical Analyses

All statistical analyses were performed in R (v.4.1) (R Core Team, 2020) using specific packages for specific analyses. To reduce the effect of variation in the sequencing depths, all samples normalized the different sequencing depths to the sample with the lowest sequencing depth using the function *rrarefy*() in the vegan package [35]. The bacterial and fungal species diversity was estimated separately by calculating species richness indices (i.e., alpha diversity) and abundance-based coverage estimator (ACE). The alpha diversity of microbial communities was compared separately for bacteria and fungi by performing one-way analysis of variance (ANOVA). Significant ANOVA results (*p* = 0.05) were subjected to post hoc multiple comparisons of means using Tukey’s test. The principal coordinate analysis (PCoA) was used to compare the bacterial and fungal community composition across the geographic locations (β-diversity) based on the Manhattan distance matrix. Differences in the overall community composition across geographical regions for bacteria and fungi were examined using analysis of similarities (ANOSIM). The function “adonis” was used to perform ANOSIM in R. A distance-based redundancy analysis (RDA) was performed to assess the relationships between environmental factors and microbial community structures. Pearson’s correlation using the Mantel test was used to determine the correlations between soil chemical properties and bacterial and fungal communities. The linear discriminant analysis (LDA) effect size (LEfSe) [36] method was used to identify significant enriched taxonomic units among different geographically distributed hull-less barley. A distance-based redundancy analysis (RDA) was employed to assess the influence of soil chemical properties on the bacterial and fungal communities’ structure, respectively [35]. Correlation between the microbial community at the phyla level and altitude and latitude and longitude profiles were tested (P_adj_  <  0.05) using Spearman’s ρ, respectively [37]. Differences in altitude, latitude, and longitude profiles and microbial communities in different samples were tested by two-way ANOVA and repeated measurements with the Bonferroni post hoc test. The significantly correlated bacterial or fungal phyla and altitude, latitude, and longitude were visualized by heatmap graphs.

### 2.5. Random Forest Classification and Prediction

Random forest (RF) machine learning classification was employed for the biomarker identification of bacteria and fungi in different geographic locations [29,38]. The relative abundances of bacterial and fungal taxa at phylum, class, order, family, genus, and ASV levels were tested separately to acquire the best-discriminated biomarkers between different geographic locations. For the prediction of different taxonomic levels, half of the hull-less barley samples were randomly selected to generate the prediction model, and the remaining hull-less barley samples were chosen to calculate the accuracies of the predictions. The most important bacterial and fungal features were also identified and screened using RF function (ntree = 10,000) and Mean Decrease Accuracy and Mean Decrease Gini, respectively. The error rates of predictions were tested by the rfcv () function (five repeats and 1000 iterations) [38].

### 2.6. Phylogenetic Tree of Core Microbial Taxa

The bacterial and fungal ASVs shared by more than 90% of rhizosphere samples were recognized as the core microbial taxa. Across all the rhizosphere samples, 255 bacterial ASVs and 117 fungal ASVs were identified as the core taxa. The ASV-associated representative sequences were extracted and used to construct maximum-likelihood phylogenetic trees (1000 replicates to perform the SH-like approximate likelihood ratio test and 1000 ultrafast bootstrap replicates) using the IQ-TREE 1.0 [39]. The core bacterial and fungal ASVs used to construct phylogenetic trees are listed in Appendix A. The representative sequences of bacterial and fungal ASVs were aligned using MUSCLE [40]. The aligned sequences were then trimmed using trimAL with the default settings [41]. The phylogenetic models for bacterial and fungal core ASVs were selected using the ModelFinder [39]. The best-fitted model using the Bayesian information criterion (BIC) was used to construct bacterial and fungal core ASVs phylogenetic trees [39]. The phylogenetic trees were visualized and annotated based on the relative abundance of each core taxa using the online open-source tool the Interactive Tree of Life (iTOL) [42].

## 3. Results

### 3.1. Rhizosphere Microbial Community Composition of Tibetan Hull-Less Barley

After low-quality sequencing data were removed, 2,371,309 high-quality fungal reads (47,426 reads per sample) and 3,094,746 high-quality bacterial reads (61,895 reads per sample) were retained. The bacterial and fungal reads were assembled into 19,424 ASVs and 3415 ASVs, respectively, using the UNOISE algorithm (100% sequence identity threshold). Rarefaction curves showed that both bacterial and fungal communities nearly reached the asymptote, indicating that the sequencing depths for rhizosphere samples were sufficient (Appendix A). Diversity analyses were performed separately for fungal and bacterial communities after the sequencing depth was normalized to the lowest level (39,860 reads for fungi and 30,111 reads for bacteria). The bacterial ASVs were annotated to 37 phyla, 95 classes, 203 orders, 277 families, and 702 genera. At the phylum level and across the samples, Acidobacteriota (24.8%), Bacteroidota (13.3%), Chloroflexi (12.2%), and Actinobacteriota (5.18%) were the most dominant phyla (Figure 1a). The fungal ASVs were annotated to 37 phyla, 95 classes, 203 orders, 277 families, and 702 genera. Ascomycota (39.7%), Mortierellomycota (15.9%), Basidiomycota (18.0%), and Chytridiomycota (9.8%) were the most dominant phyla (Figure 1c). About 21.6% of the fungal sequences were not assigned to any phylum, indicating that the Tibetan hull-less barley may have an association with many unidentified (“novel”) fungal taxa.

### 3.2. Rhizosphere Microbial Diversity and Community Structure of Tibetan Hull-Less Barley

The Manhattan distance was used to calculate the rhizosphere microbial community composition of hull-less barley using normalized sequences. The principal coordinate analysis (PCoA) results for the bacterial community showed that the first two axes explained 37.03% of the variation: PCo1 and PCo2 explained 23.25% and 13.78% of the total variation, respectively (Figure 1b). Bacterial communities were clustered according to their geographic locations: Gongga, Kanuo, Linzhou, Pulan, Bayi, and Bomi were grouped along axes 1 and 2 (Figure 1b and Appendix A). Permutational multivariate analysis of variance (PERMANOVA) tests comparing rhizosphere bacterial community composition across geographic locations showed significant differences (*p* < 0.001). Similar trends were found in the fungal communities. The PCoA results revealed that the first two axes explained 38.73% of the variation: PCo1 and PCo2 explained 24.87% and 13.86% of the total variation, respectively (Figure 1d). PERMANOVA results showed significant differences in rhizosphere fungal communities across geographical locations (*p* < 0.001), and the communities were distinctly grouped along axes 1 and 2: Gongga, Kanuo, Linzhou, Bomi, Pulan, and Kanuo (Figure 1d and Appendix A). Alpha diversity for bacterial communities showed a significant difference across geographic locations (Tukey’s test, *p* < 0.05) (Figure 2a). We found a significant negative relationship between alpha diversity and altitude (R^2^ = 0.095; *p* = 0.03) and ACE index and altitude (R^2^ = 0.031; *p* = 0.05) at each geographical location (Figure 2b,c). For fungal communities, there was a significant difference in alpha diversity across geographic locations, and Bayi has the highest alpha diversity (Tukey’s test, *p* < 0.05) (Figure 2d). Fungal alpha diversity (R^2^ = 0.039; *p* = 0.031) and ACE index (R^2^ = 0.059; *p* = 0.05) showed a significant negative correlation with altitude at each geographical location (Figure 2b,c). These results indicate that altitude is the major factor determining rhizosphere microbial diversity of hull-less barley in the Tibetan Plateau. Moreover, approximately 40.0% and 63.5% bacterial and fungal reads could not be assigned to the genus level, respectively (Figure 1b), indicating large unclassified taxa in the rhizosphere of Tibetan hull-less barley (Appendix A).

### 3.3. Core and Enriched Rhizosphere Bacterial and Fungal ASVs 

We identified the overall core rhizosphere microbial taxa (present in more than 90% of the samples) of hull-less barley. The core taxa included 255 bacterial ASVs and 117 fungal ASVs (Figure 3a,b). The most abundant core bacterial family comprised Azospirillaceae, Bacillaceae, Blastocatellaceae, Comamonadaceae, Dongiaceae, Gemmatimonadaceae, Sphingomonadaceae, Planococcaceae, Pyrinomonadaceae, Micrococcaceae, Nitrospiraceae, Peptostreptococcaceae, and Xanthobacteraceae (Figure 3a). Among fungal communities, Apiosporaceae, Bionectriaceae, Bulleribasidiaceae, Ceratobasidiaceae, Chaetomiaceae, Cladosporiaceae, Didymellaceae, Entolomataceae, Filobasidiaceae, Leptosphaeriaceae, Microascaceae, Mortierellaceae, and Nectriaceae were the most common core fungal families (Figure 3b). A linear discriminant analysis (LDA) effect size (LEfSe) method was used to identify specific and significant taxonomic units among microbial taxa and their distributions. The results suggest that barley at higher elevations was enriched with the bacteria belonging to *Acidibacter*, *Arthrobacter*, *Altererythrobacter*, *Flavitalea*, Phycisphaeraceae, *Pontibacter*, *Methylotenera*, *Microcoleus*, *Microvirga*, *Nordella*, *Rhodomicrobium*, *Tumebacillus*, *Turicibacter*, *Stenotrophobacter*, *Steroidobacter*, and *Sphingomonas* (Figure 4a and Appendix A). However, at lower elevations, hull-less barley was enriched with *Aridibacter*, *Acidovorax*, *Bradyrhizobium*, *Ramlibacter*, *Pedomicrobium*, *Pseudarthrobacter*, *Romboutsia*, *Lysinibacillus*, and *Nitrospira*. For the fungal communities, we found that fungal taxa belonging to *Alogomyces*, *Apiosporaceae*, *Arthrinium*, *Beauveria*, *Clonostachys*, *Cephaliophora*, *Deconica*, *Fusarium*, *Kernia*, *Gliomastix*, *Lepiota*, *Leptosphaeria*, *Ophiosphaerella*, *Podospora*, *Paraphoma*, *Preussia*, *Pseudogymnoascus*, *Rhizophlyctis*, *Rhizoctonia*, *Syncephalis*, *Vishniacozyma*, and *Volutella* were enriched in barley rhizosphere at high altitude (Figure 4b and Appendix A). Across low elevations, barley was substantially enriched with *Aleuria*, *Articulospora*, *Chalara*, *Exophiala*, *Humicola*, *Ganoderma*, *Phaeohelotium*, *Pseudaleuria*, *Minimelanolocus*, *Minimedusa*, *Naganishia*, *Sistotrema*, *Spegazzinia*, *Solicoccozyma*, and *Hanatephorus*. (Figure 4b and Appendix A).

### 3.4. Environmental Factors Shaping Rhizosphere Microbiota and the Biomarkers Distinguishing Different Groups 

Distance-based redundancy analysis (RDA) was conducted to characterize the significant environmental factors structuring barley’s rhizosphere bacterial and fungal communities. Selected environmental factors explained approximately 13.01% and 34.20% of the total variance in the bacterial and fungal communities, respectively (Figure 4c,d). Results suggest that altitude (*p* = 0.001), total carbon (*p* = 0.001), total nitrogen (*p* = 0.001), and available phosphorus (*p* = 0.05) and potassium ions (*p* = 0.001) were the most important factors structuring the composition of rhizosphere bacterial communities of hull-less barley (Figure 4c). We also found that altitude (*p* = 0.001), total carbon (*p* = 0.001), total nitrate (*p* = 0.001), and available potassium (*p* = 0.001) were the significant factors shaping the fungal communities (Figure 4d). 

Using bacterial and fungal communities as biomarkers, we also built the random forest (RF) model to distinguish hull-less barley from different geographical locations (Figure 5a,c). First, RF models of both bacterial and fungal communities were established at the phylum, class, order, family, and genus levels. Then, half of the samples (*n* = 25) were utilized as training data. RF models indicate that bacterial and fungal taxa at the genus level most clearly discriminated the samples from different geographic locations. The following analysis of bacterial and fungal communities was performed at the genus level. The established models for both bacteria and fungi were identified using ten-fold cross-validation with five repeats. The RF models showed that the hull-less barley-associated bacteria exhibited 98% classification accuracy, with 28 most relevant bacterial genera selected as the biomarkers (Figure 5a,b). In contrast, fungi exhibited 96% classification accuracy, and the 75 most relevant fungal genera were selected as the biomarkers (Figure 5c,d). These results indicate that the selected bacterial and fungal biomarkers were sufficient to distinguish hull-less barley from different locations. Therefore, both bacterial and fungal biomarkers can reasonably predict the original geographic location of hull-less barley.

## 4. Discussion

Hull-less barley is an important staple food in the Tibetan Plateau. Hull-less barley-associated microbiota are known to influence the health and environmental stress resistance of host plants [19]. However, we have limited knowledge about rhizosphere microbial communities of hull-less barley. Since rhizosphere microbial communities play essential roles in plant growth, disease resistance, nutrient cycling, and ecosystem sustainability [43,44,45], a better understanding of the distribution and structure of hull-less barley-associated microbiota may help to establish positive plant–microbe interactions and improve crop yield [22]. Our results showed that rhizosphere bacterial and fungal communities of hull-less barley were significantly and negatively correlated with the altitude. Correlation analysis showed significant positive correlations between altitude and several bacterial phyla, Chytridiomycota, Zoopagomycota, Olpidiomycota, Gemmatimonadota, Patescibacteria, Planctomycetota, WS2, and Halanaerobiaeota (|R| > 0.70, *p* < 0.05), from different geographic locations. However, Nitrospirota, Entotheonellaeota, Methylomirabilot, and Basidiomycota showed a significant negative correlation with altitude (Appendix A). In Tibetan hull-less barley, microbial communities, specifically rhizosphere microorganisms, play an even more important role in crop growth, nutrient absorption, and disease resistance [19]. Moreover, bacterial communities are more diverse than fungal communities, and bacteria can more robustly maintain ecological plateau farmland systems [19]. The host genotype of barley could affect the endophytic and rhizosphere microbial community, driving changes in the microbial community in the rhizosphere of hull-less barley [5,42].

Geography and altitude have been described as important factors determining the distribution and structure of plant and microbial communities [46,47]. Consistent with this statement, our results also showed significant negative correlations between bacterial and fungal diversities and altitude, suggesting that the microbial communities likely perform better at low elevations. Reduced microbial community diversity at higher elevations could be attributed to harsh environmental conditions such as lower temperatures, intense radiation, and low oxygen levels [48]. However, low microbial diversity at high elevations implies that rhizosphere microbial communities of hull-less barley might not provide enough support for plants to adapt to harsh environmental conditions and protection against diseases in the Tibetan Plateau. Nevertheless, previous studies have suggested that plant-associated microbial communities help host plants to adapt to extreme environmental conditions [49]. For example, endophytic microbial communities may produce plant growth hormones (e.g., auxins and cytokinins) [50] that improve host plant tolerance to drought and other abiotic stresses [51]. In addition, endophytic microbial communities provide plants with easy access to limited nutrients such as phosphorus and potassium [52,53]. Despite having low diversity, we found specific bacterial and fungal taxa enriched at high altitudes, and these microbiota may still provide enough support for the survival, growth, and development of hull-less barley [19]. 

Tibetan Plateau is known as “the third pole of the earth” and is a unique place on the planet [5]. Due to its remote geographical location, difficult topography, high elevation, and harsh environmental conditions, there are many unexplored organisms, including numerous “novel” bacterial and fungal taxa [54]. Consistent with the previous study (Liu et al., 2019), we found the fungal taxa of hull-less barley from the Qinghai-Tibet Plateau cannot be fully deciphered. Although the SILVA and UNITE databaseS are the most comprehensive database for fungal species worldwide [31], we could not assign generic names for more than 40.0% of bacterial (Appendix A) and 63.5% of fungal taxa (Appendix A). We argue that these unidentified “novel” microbes may play a significant role in plant growth as well as protection against diseases [13,55,56]. Our taxonomic annotation results also provide some evidence that the Tibetan Plateau may contain many unidentified bacterial and fungal taxa that need further exploration. These “novel” microbial taxa may serve as valuable resources for discovering new drugs and other important secondary metabolites [57]. We also suggest that future studies should aim to isolate, culture, and cultivate novel taxa that interact with hull-less barley, provide support in harsh environmental conditions, and increase yield [58]. Basic information about the altitudinal and geographical distribution of hull-less barley and their rhizosphere microbiota reported here should provide a strong basis for future studies that may unravel the association between important crops and their rhizosphere microbial communities in the Tibetan Plateau.

## 5. Conclusions

This study revealed the relationship between the microbiota of hull-less barley and the association with the altitudinal gradient and evaluated the influencing factors that shape the hull-less barley-associated bacterial and fungal compositions. We found that both bacterial and fungal diversities were significantly negatively correlated with altitude, and they were significantly affected by environmental factors such as total carbon, total nitrogen, and available phosphorus and potassium. We also built random forest models for both bacterial and fungal communities of hull-less barley, which could readily predict the original source of imported soybean grains with accuracies greater than 90%. In addition, the core bacterial and fungal taxa of Tibetan hull-less barley, as well as the geographic specific microbial taxa, were also identified. These core and specific microbial taxa may play important roles in the adaptation of the hull-less barley to the earth’s third pole.

## Figures and Tables

**Figure 1 microorganisms-10-01737-f001:**
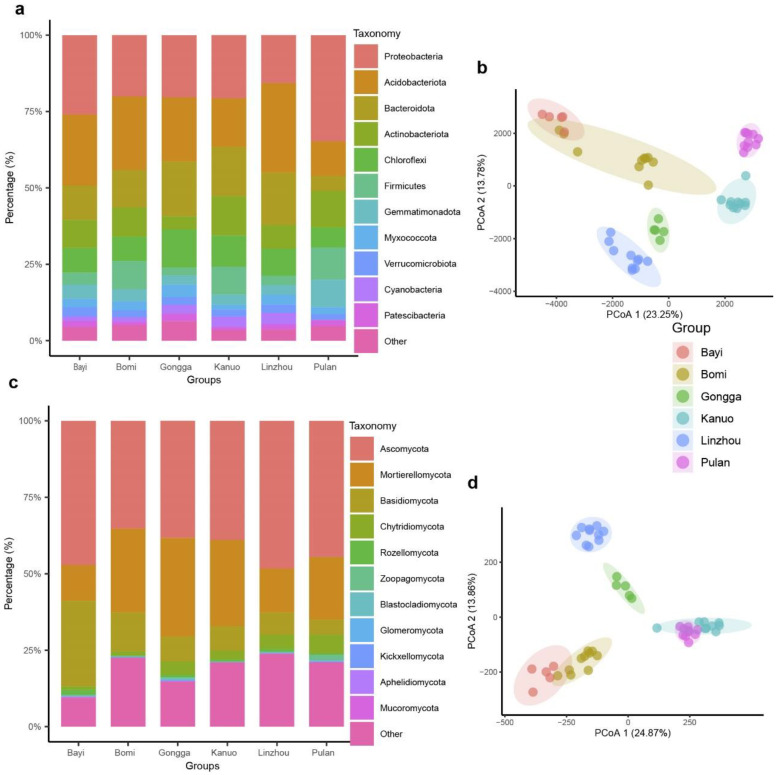
(**a**) Relative abundance of the most dominant bacterial phyla from six different geographical locations, (**b**) the principal coordinate analysis (PCA) of bacterial communities from six different locations showing PCo1 and PCo2, (**c**) relative abundance of the most dominant fungal phyla from six different geographic locations, and (**d**) PCoA of hull-less barley fungal communities from six different locations showing PCo1 and PCo2.

**Figure 2 microorganisms-10-01737-f002:**
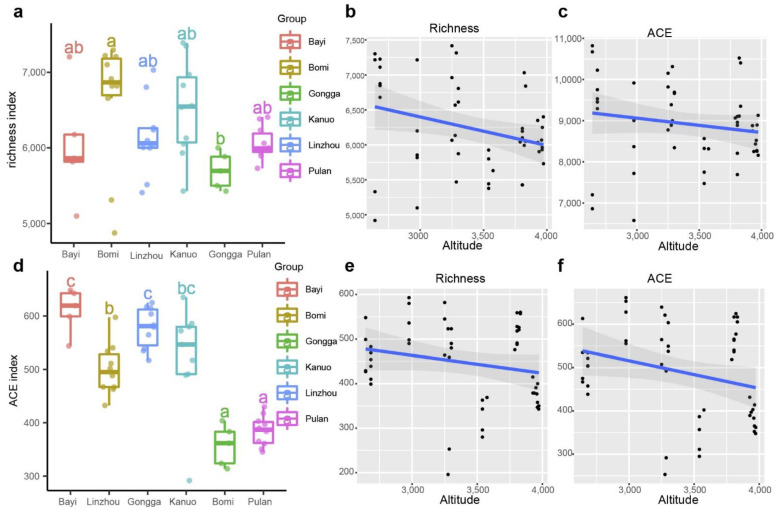
Alpha diversity and ACE index of rhizosphere microbial communities of hull-less barley at different geographical locations and their relationships with altitude: (**a**) bacterial alpha diversity across geographical locations, (**b**) correlation between bacterial alpha diversity and altitude (R^2^ = 0.095; *p* = 0.03), (**c**) correlation between ACE index of bacterial community and altitude (R^2^ = 0.031; *p* = 0.05), (**d**) fungal alpha diversity across geographical locations, (**e**) correlation between fungal alpha diversity and altitude (R^2^ = 0.039; *p* = 0.031), and (**f**) correlation between the ACE index of fungal community and altitude (R^2^ = 0.059; *p* = 0.05). For (**a**,**d**) bars with different letters represent significant differences across geographical locations (α = 0.05).

**Figure 3 microorganisms-10-01737-f003:**
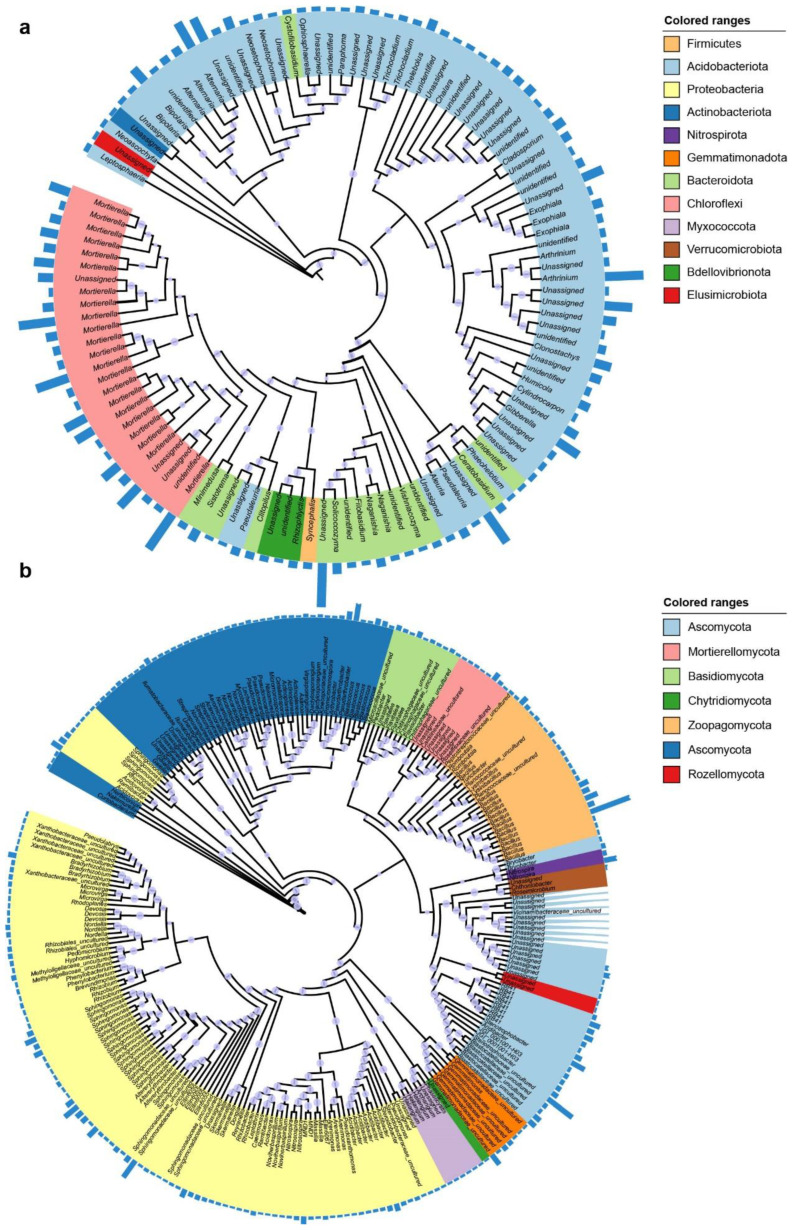
Phylogenetic relationships among core (**a**) bacterial ASVs and (**b**) fungal ASVs (present in >90% of the rhizosphere samples). The blue bar plots (a represent bacteria and b represent fungi) represent the relative abundance of core ASVs. Each node of the phylogenetic tree was colored by the phylum.

**Figure 4 microorganisms-10-01737-f004:**
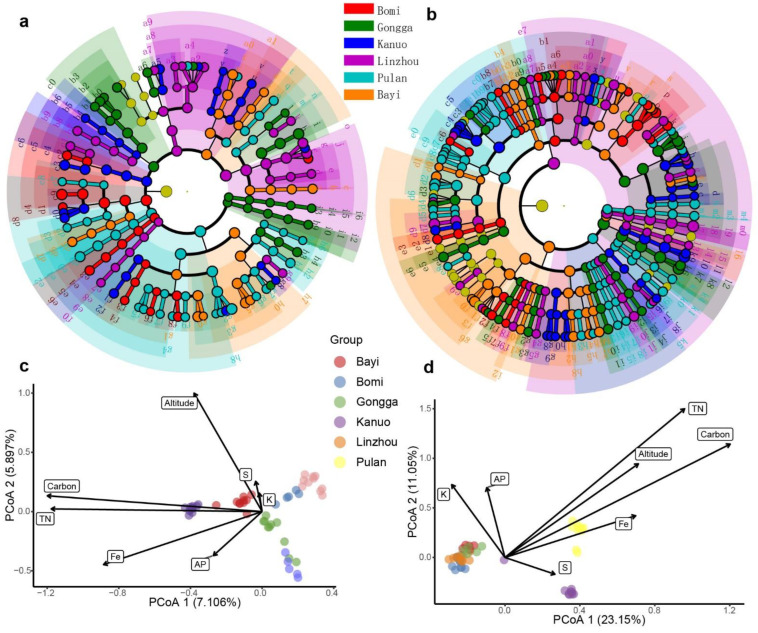
(**a**) A linear discriminant analysis (LDA) effect size (LEFSe) taxonomic cladogram comparing the best-discriminated bacterial taxa representing different geographical locations. Significantly discriminant taxon nodes are colored according to the highest-ranked group for that taxon and (**b**) LDA effect size (LEFSe) taxonomic cladogram comparing the best-discriminated fungal taxa representing different geographic locations. Significantly discriminant taxon nodes are colored according to the highest-ranked group for that taxon. A distance-based redundancy analysis (RDA) diagram depicting (**c**) bacterial communities of barley in relation to various measured environmental variables from six different geographical locations and (**d**) fungal communities of barley in relation to various measured environmental variables.

**Figure 5 microorganisms-10-01737-f005:**
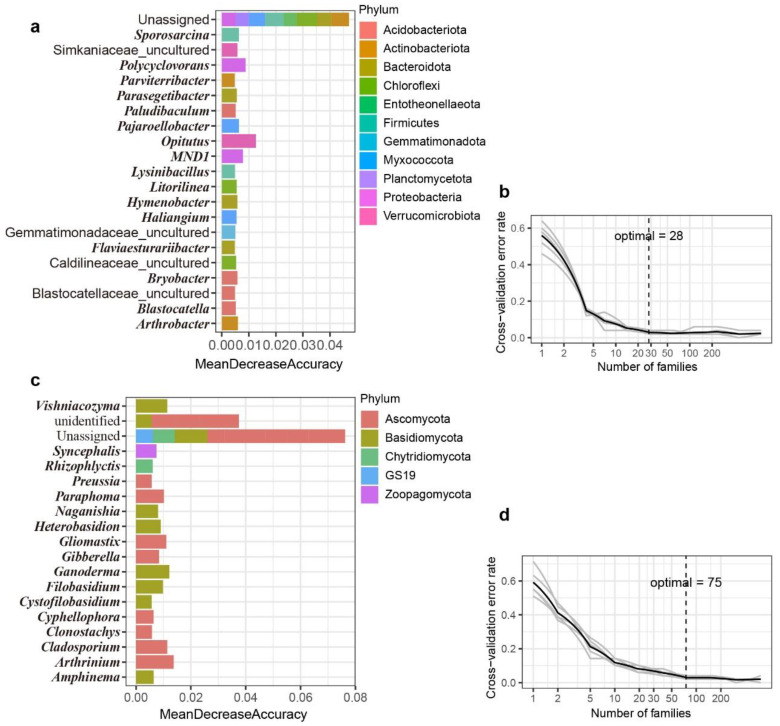
Random forest models for bacterial and fungal communities built at the generic levels show the lowest classification errors. (**a**) Twenty-eight bacterial biomarkers were selected by random forest classification and relative abundance, and (**c**) seventy-five fungal biomarkers were selected by random forest classification and relative abundance. The bar plot of each taxon indicates the taxa’s importance to the model accuracy. The ten-fold cross-validation error and identified number of (**b**) bacterial (**b**,**d**) fungal biomarkers were used to differentiate hull-less barley from different geographical locations.

## Data Availability

The raw read data have been deposited into the National Center for Biotechnology Information (NCBI) Sequence Read Archive database under BioProject accession number PRJNA856343.

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
