# Peer review of "Distribution of Core Root Microbiota of Tibetan Hulless Barley along an Altitudinal and Geographical Gradient in the Tibetan Plateau"

_microorganisms, 2022, doi:10.3390/microorganisms10091737_

Round 1

Reviewer 1 Report

The manuscript is in a greata form, and im my opnion should be accept for publication. The authors exploit the data, to acquire the answerers to the hypothesis mentioned. Further they used the corrects method e methodology, used high level graphs and obtained conclusions the may driveothers new questions that will move forward the scientific mean about the Distribuition of the microbial communities associated to the development of plants and then shake the knowledge of how plants can be modulate in its development by microbes and whats are the clues to increasing the prodution of food.

Therefore I should congratulate the authors and indicate this manuscript for publication.

Author Response

We appreciate the compliments from reviewer on our presentation style of figures, and other aspects. We hope our findings from HTS studies could contribute to the understanding of the microbial community of Tibetan hull-less barley.

Reviewer 2 Report

1. Authors are suggested to write the criteria for choosing the sampling sites.

2. Image quality is not up to the mark. Authors are suggested to replace these images with good-quality images.

3. Mention the depth where the soil samples were collected from.

4. There are a few grammatical and syntactical errors within the manuscript. Authors are suggested to remove such errors to make the article effective and informative.

5. Write a few lines about the importance of the current study along with the future possibilities.

6. Authors are suggested to write a paragraph of the conclusion just after the discussion section. It must contain the core findings of the study.

Author Response

Dear Editor and Reviewers,

  Submission of revised manuscript

Thank you indeed for considering our manuscript No. 1846073 entitled “Distribution of core root microbiota of Tibetan hull-less barley along an altitudinal and geographical gradient in the Tibetan Plateau”. Your comments and suggestions are valuable and helpful for improving our manuscript. We have revised our manuscript according to reviewers’ suggestions. Please find below a point-by-point response to each reviewer’s comment. We hope that it is now acceptable for publication in Microorganisms.

On behalf of my co-authors, I would like to express our appreciation to you for the great help in improving the manuscript.

Yours Sincerely

Na Wei

on behalf of all authors

Comments and Suggestions for Authors

  1. Authors are suggested to write the criteria for choosing the sampling sites.

R:Thank you for your suggestion. We have added the criteria for choosing the sampling sites in Line108-111.

  1. Image quality is not up to the mark. Authors are suggested to replace these images with good-quality images.

R:Thank you for your suggestions, we have re-uploaded high-quality format figures (300 ppi) to replace the old one. They will be sufficient to meet the requirement for publication.

  1. Mention the depth where the soil samples were collected from.

R:Thank you. We have added the detailed sample collection methods in M&M in Line 105-114. Approximately 5 cm of topsoil was removed, and the depth of soil samples we collected are around 5cm ~15 cm.

  1. There are a few grammatical and syntactical errors within the manuscript. Authors are suggested to remove such errors to make the article effective and informative.

R:We have carefully checked the full text as suggested, and the grammatical and syntactical errors have been revised. We hope the overall readability of our revised manuscript will be better.

  1. Write a few lines about the importance of the current study along with the future possibilities.

R:Thank you for your suggestions. We have added several sentences to introduce the importance of the current study in Line 407-418.

  1. Authors are suggested to write a paragraph of the conclusion just after the discussion section. It must contain the core findings of the study.

R:Thank you for your suggestions. We have added a paragraph of the conclusion part after the discussion section. We summarized the core findings in the conclusion part.

Reviewer 3 Report

The paper entitled “Distribution of core root microbiota of Tibetan hull-less barley along an altitudinal and geographical gradient in the Tibetan Plateau” is an original article. The article will contribute to science as well. The study focused on the composition and diversity of bacterial and fungal communities associated with the hull-less barley at various elevations in the Tibetan Plateau.

I think there is the nucleus of a paper here. However, I think considerably more work is needed in: (1) the writing and presentation of the manuscript as a whole, (2) the description and analysis of results, and (3) the overall framing of the work.

The abstract is well written and provides the important points of the articles.

Keywords have repetitions from the title that needs to be revised.   

The introduction needs to include one more paragraphs that correlate the current objectives with the previous literature.

The discussion needs more comparative studies to be included in.

The figures need to be more clear and high quality

References need to be formatted in good order, according to the journal format.

Author Response

Dear Editor and Reviewers,

  Submission of revised manuscript

Thank you indeed for considering our manuscript No. 1846073 entitled “Distribution of core root microbiota of Tibetan hull-less barley along an altitudinal and geographical gradient in the Tibetan Plateau”. Your comments and suggestions are valuable and helpful for improving our manuscript. We have revised our manuscript according to reviewers’ suggestions. Please find below a point-by-point response to each reviewer’s comment. We hope that it is now acceptable for publication in Microorganisms.

On behalf of my co-authors, I would like to express our appreciation to you for the great help in improving the manuscript.

Yours Sincerely

Na Wei

on behalf of all authors

The paper entitled “Distribution of core root microbiota of Tibetan hull-less barley along an altitudinal and geographical gradient in the Tibetan Plateau” is an original article. The article will contribute to science as well. The study focused on the composition and diversity of bacterial and fungal communities associated with the hull-less barley at various elevations in the Tibetan Plateau.

I think there is the nucleus of a paper here. However, I think considerably more work is needed in: (1) the writing and presentation of the manuscript as a whole, (2) the description and analysis of results, and (3) the overall framing of the work.

R:We thank Reviewer#3 for the positive feedback. We have carefully checked and revised the full text to increase the overall readability of the manuscript. We also added some paragraphs in the introduction and discussion parts and revised part of the analysis of results to increase the overall framing of the manuscript.

The abstract is well written and provides the important points of the articles.

R:Thanks.

Keywords have repetitions from the title that needs to be revised.

R: Thank you for your suggestion. We have deleted the repeated keywords and added new keywords in Line32-35.

The introduction needs to include one more paragraphs that correlate the current objectives with the previous literature.

R:Thank you for your suggestions. Based on the searching of web of science core collection database (www.webofknowledge.com) with key words “Tibetan barley” and “rhizosphere”, only four references were available in recent five years. This result indicate studies about the rhizosphere and microorganisms in Tibetan are lacking, and more researches should be conducted to explore the Tibetan hulless barley-associated microbiome. We have added one paragraph in the introduction part to introduce the correlation of current objectives with these previous four literatures.

The discussion needs more comparative studies to be included in.

R:Thank you for your suggestions. We have added comparative studies and the importance of the current study along with the future possibilities in the discussion part.

The figures need to be more clear and high quality

R:Thank you for your suggestions. We have re-uploaded high-quality format figures (300 ppi) to replace the old ones. They will be sufficient to meet the requirement for publication.

References need to be formatted in good order, according to the journal format.

R: Appreciated. We have reformatted all the references according to the journal’s recommendation.